# Quantitative Risk Assessment of Susceptible and Ciprofloxacin-Resistant *Salmonella* from Retail Pork in Chiang Mai Province in Northern Thailand

**DOI:** 10.3390/foods11192942

**Published:** 2022-09-20

**Authors:** Chaiwat Pulsrikarn, Anusak Kedsin, Parichart Boueroy, Peechanika Chopjitt, Rujirat Hatrongjit, Piyarat Chansiripornchai, Nipattra Suanpairintr, Suphachai Nuanualsuwan

**Affiliations:** 1National Institute of Health, Department of Medical Science, Ministry of Public Health, Nonthaburi 11000, Thailand; 2Faculty of Public Health, Kasetsart University, Chalermphrakiat Sakon Nakhon Province Campus, Sakon Nakhon 47000, Thailand; 3Faculty of Science and Engineering, Chalermphrakiat Sakon Nakhon Province Campus, Kasetsart University, Sakon Nakhon 47000, Thailand; 4Department of Pharmacology, Faculty of Veterinary Science, Chulalongkorn University, Bangkok 10330, Thailand; 5Center of Excellence for Food and Water Risk Analysis (FAWRA), Department of Veterinary Public Health, Faculty of Veterinary Science, Chulalongkorn University, Bangkok 10330, Thailand; 6Department of Veterinary Public Health, Faculty of Veterinary Sciences, Chulalongkorn University, Bangkok 10330, Thailand

**Keywords:** AMR, probabilistic models, quinolone, retail pork, risk assessment, *Salmonella*

## Abstract

The adverse human health effects as a result of antimicrobial resistance have been recognized worldwide. *Salmonella* is a leading cause of foodborne illnesses while antimicrobial resistant (AMR) *Salmonella* has been isolated from foods of animal origin. The quantitative risk assessment (RA) as part of the guidelines for the risk analysis of foodborne antimicrobial resistance was issued by the Codex Alimentarius Commission more than a decade ago. However, only two risk assessments reported the human health effects of AMR *Salmonella* in dry-cured pork sausage and pork mince. Therefore, the objective of this study was to quantitatively evaluate the adverse health effects attributable to consuming retail pork contaminated with *Salmonella* using risk assessment models. The sampling frame covered pork at the fresh market (*n* = 100) and modern trade where pork is refrigerated (*n* = 50) in Chiang Mai province in northern Thailand. The predictive microbiology models were used in the steps where data were lacking. Susceptible and quinolone-resistant (QR) *Salmonella* were determined by antimicrobial susceptibility testing and the presence of AMR genes. The probability of mortality conditional to foodborne illness by susceptible *Salmonella* was modeled as the hazard characterization of susceptible and QR *Salmonella.* For QR *Salmonella*, the probabilistic prevalences from the fresh market and modern trade were 28.4 and 1.9%, respectively; the mean concentrations from the fresh market and modern trade were 346 and 0.02 colony forming units/g, respectively. The probability of illness (*P*_I_) and probability of mortality given illness (*P*_MI_) from QR *Salmonella*-contaminated pork at retails in Chiang Mai province were in the range of 2.2 × 10^−8^–3.1 × 10^−4^ and 3.9 × 10^−10^–5.4 × 10^−6^, respectively, while those from susceptible *Salmonella* contaminated-pork at retails were in the range 1.8 × 10^−4^–3.2 × 10^−4^ and 2.3 × 10^−7^–4.2 × 10^−7^, respectively. After 1000 iterations of Monte Carlo simulations of the risk assessment models, the annual mortality rates for QR salmonellosis simulated by the risk assessment models were in the range of 0–32, which is in line with the AMR adverse health effects previously reported. Therefore, the risk assessment models used in both exposure assessment and hazard characterization were applicable to evaluate the adverse health effects of AMR *Salmonella* spp. in Thailand.

## 1. Introduction

The adverse health effects posed by antimicrobial-resistant (AMR) bacteria have been increasing at an alarming rate and have recently been recognized worldwide [1]. Antimicrobial agents are beneficial to a wide variety of sectors, from human and veterinary medicine to animal and plant production. The use of antimicrobial agents in these areas is inevitably connected and this renders a circulating pool of both resistant bacteria and bacteria-borne resistant genes that are eventually delivered to humans [2]. Regardless of the environmental, genetic, or spatial boundary, mobile genetic elements containing resistance determinants can, directly and indirectly, propagate through horizontal transfer among bacteria from foods of animal origin and their environment to humans. Therefore, AMR risk management measures in terms of prevention and control strategy rely heavily on source attribution and risk assessment to evaluate the likelihood and severity of the consequences of AMR bacteria-contaminated foods [3].

The Codex Alimentarius Commission (CAC) has endorsed a systematic framework for foodborne AMR risk analysis. This framework is composed of preliminary risk management activities, risk assessment, and risk management. These three components are connected via risk communication, including the surveillance of AMR and other sources of information. The underlying rationale of the principle of risk analysis is to evaluate the risk to human health from foodborne AMR microorganisms and AMR determinants so that practical risk management measures can be implemented to prevent and control such human health risks [4].

The microbial risk assessment is a scientific process to evaluate the risk of consuming food contaminated with hazards. Hazard identification, hazard characterization, exposure assessment, and risk characterization constitute the four-step risk assessment. Hazard identification is the initial step to examining the risk of the hazards such as foodborne disease viruses, bacteria, protozoa, and parasites; in this study, the hazard is *Salmonella*. Hazard characterization determines the probability of illness upon getting a hazard into the host by a specific dose–response model. The exposure assessment determines the probability of getting hazards through consuming food. The last step is risk characterization, where the risk estimate is derived from the product of probabilities of exposure and illness from the preceding two steps [5,6]. Foodborne AMR risk assessment (AMR RA) is slightly different from the traditional methodology of microbiological risk assessment [5,6] in that hazard characterization is necessary to additionally include the adverse effects of AMR, e.g., antimicrobial treatment failure, prolonged treatment period, more illness severity or virulence, and higher mortality rate [4].

The prevalence of susceptible *Salmonella* spp. from swine manure was in the range of 2–61% and from swine farm swabs it was 95%, whereas that of AMR *Salmonella* spp. isolated from antimicrobial-use swine farms was lowest at 33% against florfenicol and highest at 66% against tetracycline [7]. Likewise, the prevalence of tetracycline-resistant *Salmonella* spp. was even higher at 90% in two independent studies [8,9]. However, isolated *Salmonella* spp. was sensitive to ceftiofur, ceftriaxone, and ciprofloxacin. In addition, fluoroquinolone-resistant *Salmonella* spp. warrant further surveillance by the World Health Organization as a tier group 2 [10].

AMR *Salmonella* spp. in retail pork could be derived either from a farm or abattoir. The prevalence of AMR *Salmonella* spp. from the environment of the abattoir was lowest at 4% against ceftiofur and highest at 86–89% against tetracycline [7,8]. Recently, we investigated a total of 387 non-typhoidal *Salmonella enterica* (NTS) isolated from abattoirs. Approximately 24% of NTS isolates were AMR, while only 6% of NTS isolates were susceptible to all antimicrobial agents tested. However, non-AMR NTS isolates carry extended-spectrum beta-lactamase (*bla*_CTX-M_) genes or narrow-spectrum beta-lactamase genes (*bla*_TEM_ or *bla*_SHV_). The rest of the NTS isolates (70%) were susceptible to all fluoroquinolones as well as carbapenems and third-generation cephalosporins [11]. At retail, *Salmonella* spp. isolated from pork was susceptible to ampicillin, norfloxacin, and ciprofloxacin [8,12]. However, the prevalence of AMR *Salmonella* spp. isolated from retail pork were 100% against streptomycin and sulfamethoxazole [12] and 60% against tetracycline [8]. The source attribution of salmonellosis in children from pork was 11% [8].

Scientific evidence demonstrated that AMR *Salmonella* spp. is foodborne-transmitted. On the other hand, AMR *Salmonella* infection is seldom traced from patients in hospitals back through contaminated foods and even further back to animals along the food chain. Some of the implicated commodities in such reports were beef, pork, and milk, where the authors suggested that AMR *Salmonella* spp. in patients was attributable to farm animals [13,14,15]. While commonly found *Salmonella* serovars with either resistance or multiple resistance to *Salmonella* spp. in foods are Derby, Enteritidis, Hadar, Newport, Paratyphi, Typhimurium, and Virchow [16,17,18], *Salmonella* Typhimurium is the most prevalent serovar contaminating foods across continents [13,14,15,17,19,20]. Recently, both cephalosporin-resistant and extended-spectrum beta-lactamase-resistant *Salmonella* spp. have been frequently reported [21,22]. These reports implied that AMR *Salmonella* spp. has been widely circulated regardless of geographical borders, food commodities, serovars, resistance patterns, antimicrobial classes, and host ranges.

Even though 34 AMR RA relating to retail foods have been reported up to 2018, only eight articles investigated the adverse health effects of AMR *Salmonella* spp. Only half of these reports are related to the pork supply chain [23]. Two risk assessments reported the adverse health effects of AMR *Salmonella* in dry-cured pork sausage and pork mince [24,25]. Recently, a farm-to-fork quantitative risk assessment of *Salmonella* Heidelberg resistant to third-generation cephalosporins in broiler chickens was reported [26] while the AMR RA model was developed for anti-*E. coli* drugs [27]. However, a quantitative risk assessment using Monte Carlo simulation of QR *Salmonella* in retail pork has never been reported. In this study, QR *Salmonella*-contaminated pork was the hazard of interest. The sampling frame covered pork at retailers in Chiang Mai province in northern Thailand. The predictive microbiology models were used in the steps where data were lacking. The objective of this study was to comparatively evaluate the adverse health effects attributable to consuming pork contaminated with *Salmonella* susceptible and resistant to quinolone.

## 2. Materials and Methods

### 2.1. Pork Samples

The pork samples were collected in Chiang Mai province from both the fresh market and modern trade where pork is refrigerated. Ten pork samples were collected from each retailer. Eleven pork retailers from the fresh market and five butcher shops in the modern trade participated in this study. The sampling unit of pork was at least 100 g. Samples were collected using an aseptic technique to avoid undesirable cross-contamination from environmental fomite and then kept in a leak-proof container between 2 and 8 °C during transportation. The samples arrived at the laboratory and were analyzed within 8–10 h after being collected.

### 2.2. Enumeration of Salmonella

The ten-fold serial dilution of pork samples was achieved using buffered peptone water. For individual dilution, 1 mL of suspension was repeatedly transferred 3 times into 3 separate 9 mL tubes of Rappaport Vassiliades with soya (RVS) broth. Nine tubes of RVS broth for each sample were incubated at 42 °C for 24 h. Only RVS tubes with a turbid appearance and confirmed by xylose lysine desoxycholate agar and then triple sugar iron slant were counted as positive [28]. The concentrations of *Salmonella* in the Most Probable Number unit (MPN) were converted to colony-forming units (cfu) by multiplying by 0.8 since the MPN technique is more sensitive than a standard plate count by 25% [29]. The unit conversion of concentration is necessary to apply for a dose–response model using the dose unit as the cfu [30].

### 2.3. Antimicrobial Susceptibility Testing (AST)

Susceptibility testing for ampicillin, cefepime, cefotaxime, cefoxitin, chloramphenicol, ciprofloxacin, colistin, gentamicin, imipenem, meropenem, nalidixic acid, streptomycin, sulphamethoxazole, tetracycline, and trimethoprim was performed using a broth microdilution assay to determine the minimum inhibitory concentration (MIC) according to the M07 Clinical and Laboratory Standards Institute (CLSI) guidelines [31]. The results were interpreted according to the 2020 Clinical and Laboratory Standards Institute guidelines for the susceptibility testing of *Salmonella* isolates [32]. *Escherichia coli* ATCC 25922 was used as the control. The broth microdilution assay was performed using two-fold dilution at a concentration in a range of 0.03–64 μg/mL depending on the antimicrobial agents, which are suggested based on the 2020 CLSI.

### 2.4. Determination of Antimicrobial Resistance Genes

Antimicrobial resistance genes including quinolone, colistin, and carbapenem were conducted using a polymerase chain reaction (PCR). The PCR was carried out to determine the quinolone resistance determining region of *gyrA* and *parC*, and the plasmid-mediated quinolone resistance genes following are described elsewhere [33,34]. The PCR products of the quinolone resistance determining the region from the four genes were purified and subjected to Sanger sequencing (performed by Apical Scientific Sdn Bhd, Selangor, Malaysia) to determine their substitution by comparing with those of wild-type *S.* Typhimurium LT2 [11]. The presence of antibiotic resistance-conferring genes of colistin, including *mcr-1* through *mcr-9*, and carbapenem consisting of *bla*_NDM_, *bla*_OXA-48-like_, *bla*_IMP_, and *bla*_KPC_ was investigated using the PCR method described elsewhere [11]. All PCRs performed in this study are described in the Appendix A.

### 2.5. Risk Assessment Models

#### 2.5.1. Exposure Assessment

Probabilistic prevalence variable

The range of prevalence is between zero (0%) and one (100%), inclusively applicable to the range of Beta distribution. The Beta distribution is characterized by 2 parameters, alpha and beta, as shown in Equation (1).
*P*_PROB_ = Beta (*α*, *β*)(1)

To describe the variability of prevalence, the alpha parameter is substituted by *s* + *α*, and the beta parameter is substituted by *n* − *s* + *β* where *s* is the number of the successful trial (*s*) in the identical *n* trials of a binomial process, as shown in Equation (2). In this study, the successful trials were the QR *Salmonella*-contaminated (positive) samples where the identical *n* trials were the sample size.
*P*_PROB_ = Beta (*s + α*, *n* − *s + β*)(2)

This study assumes that no prior prevalence of QR *Salmonella* was reported. The uniform probability distribution was assumed, which is equivalent to Beta (1, 1). Therefore, two parameters in Equation (2) were replaced with 1, as shown in Equation (3) [6].
*P*_PROB_ = Beta (*s +* 1, *n* − *s +* 1)(3)

2.Thermal inactivation model

The raw pork from retail was subjected to heat treatment before consumption. The cooking temperature and time were 64 °C for 2 min while the decimal reduction time at 64 °C (*D*_64_) is 0.48 min [30]. The log reduction of *Salmonella* is shown in Equation (4).
(4)$$LR=\frac{t}{D_{64}}$$
where LR = log reduction (LR) of susceptible or QR *Salmonella* in pork; *D*_64_ = decimal reduction time of *Salmonella* at 64 °C (min); *t* = cooking time (min).

3.Concentration variable

If pork samples were all negative, the *Salmonella* concentration was determined by the maximum likelihood estimator (MLE) technique [29,35,36], as shown in Equation (5).
(5)$$\log\;\mathrm{reduction}\;\left(\mathrm{LR}\right)\;C_{S}=\frac{\sum_{i=1}^{k}N_{i}}{\sum_{i=1}^{k}V_{i}}-10^{LR}$$
where *C_S_* = concentration of susceptible or QR *Salmonella* (g^−1^); *N_i_* = no. of *Salmonella* detected in retail pork *i* to *k*; *V_i_* = analytical unit of pork *i* to *k* (g); *k* = no. of pork retailers; *LR* = log reduction of *Salmonella* from heat treatment.

4.Consumption variable (*C_P_*)

Food consumption data for Thailand in 2016 from the Agricultural Commodity and Food Standard report showed that the mean and 97.5th percentile consumption of pork among eaters more than 3 years old was 14.12 and 58.28 g/person/day, respectively. The triangular distribution was used to describe the variability of the consumption variable. The three parameters of triangular distribution (minimum, most likely, and maximum) were 0, 14.12, and 58.28 g/person/day, respectively.

5.Dose of *Salmonella* ingested

The dose of *Salmonella* ingested was the product of *Salmonella* concentration after cooking and pork consumption per day. The equation for the dose of *Salmonella* ingested is shown in Equation (6) [6].
*D* = *C*_S_ × *C*_P_(6)
where *D* = dose of susceptible or QR *Salmonella* ingested per day (cfu); *C*_S_ = concentration of susceptible or QR *Salmonella* (cfu/g); *C*_P_ = pork consumption per day (g).

6.Probability of exposure (*P*_E_)

*P*_E_ is the likelihood of experiencing at least one cell of *Salmonella* from pork. Therefore, the input variables to model the *P*_E_ are the concentration (*C*_S_) and prevalence (*P*_PROB_) of *Salmonella*, including pork consumption (6), as shown in Equation (7).
*P*_E_ = *P*_PROB_ (1 − exp − *D*)(7)

#### 2.5.2. Hazard Characterization

Probability of illness (*P*_I_)

The dose–response model was used to characterize the probability of illness caused by either residual susceptible or QR *Salmonella*-contaminated pork after cooking, as shown in Equation (8).
*P*_I_ = 1 − (1+ (*D*/51.45))^−0.1324^(8)
where *P*_I_ = the probability of illness caused by an ingested dose of *Salmonella*; *D* = dose of susceptible or QR *Salmonella* ingested per day (cfu).

2.Probability of mortality (*P*_M_)

Additional to the conventional hazard characterization of the microbial risk assessment, the adverse effects of AMR such as a higher mortality rate were included [4]. A previous study reported that the mortality rates caused by drug-susceptible and multidrug-resistant non-typhoid *Salmonella* were 0.2 and 3.4%, respectively [13]. Likewise, another study reported that the mortality rates caused by pan-susceptible and AMR *Salmonella* were 0.06 and 0.1%, respectively [18]. Therefore, in this study, the mean mortality rates as *P*_M_ caused by susceptible and AMR *Salmonella* were averaged from these two previous reports as 0.13 and 1.75%, respectively.

3.Probability of mortality given illness (*P*_MI_)

The integration of adverse health effects as the mortality conditional to the foodborne illness is the product of *P*_I_ and *P*_M_, as shown in Equation (9).
*P*_MI_ = *P*_M_ × *P*_I_(9)

#### 2.5.3. Risk Characterization

In this study, the risk characterization is a two-step linked process of exposure assessment and hazard characterization. The probability of mortality given illness (*P*_MI_) is conditional on *P*_E_. Assuming that adverse health effects and hazard exposure are independent, the model for risk estimates in terms of the probability of foodborne mortality (*P*_FM_) is the product of *P*_MI_ and *P*_E_, as shown in Equation (10).
*P*_FM_ = *P*_MI_ × *P*_E_(10)

The probability of foodborne mortality from at least one day was calculated based on the binomial theorem [36]. The number of annual foodborne mortality cases per 100,000 population is calculated from Equation (11).
*M*_AFM_ = (1 − (1 − *P*_FM_)^365^) × 100,000(11)
where *M*_AFM_ = annual foodborne mortality cases per 100,000 population; *P*_FM_ = probability of foodborne mortality per day.

Simulations of *M*_AFM_ were run for 10,000 iterations. The Simulación 4.0 freeware (developed by José Ricardo Varela) was used to run the Monte Carlo simulations.

### 2.6. Statistical Analysis

The *M*_AFM_ of susceptible and QR *Salmonella* in pork from the fresh market and modern trade was determined for the statistical difference by one-way analysis of variance (ANOVA) [37]. Tukey’s multiple comparison test was followed to determine the pair-wise differences of *M*_AFM._ The IBM^®^ SPSS^®^ Statistics version 22 software (SPSS Inc., Chicago, IL, USA) was used to perform statistical analyses.

## 3. Results

### 3.1. Exposure Assessment

A total of 150 pork samples collected from pork retailers (fresh market (*n* = 100) and modern trade (*n* = 50)) in Chiang Mai province were analyzed for *Salmonella* contamination. The number of *Salmonella*-positive samples is shown in Table 1. All *Salmonella* isolates from positive samples were subject to the AST. We determined antimicrobial-resistant genes in QR isolates for colistin (*mcr-1* through *mcr-9*), carbapenem, and fluoroquinolone including *mcr*, *bla*_NDM_, *bla*_OXA-48-like_, *bla*_IMP_, *bla*_KPC_, plasmid-mediated quinolone resistance, and the quinolone resistance-determining region of *gyrA* and *parC*. No isolates carried the mobile colistin resistance gene (*mcr*) and common carbapenemase genes (*bla*_NDM_, *bla*_OXA-48-like_, *bla*_IMP_, *bla*_KPC_). In the case of fluoroquinolone-resistant genes, among the QR isolates, five isolates carried *qnrS*, there were two substitutions in *parC*, and one isolate carried both *qnrS* and *parC* substitutions. No substitution occurred in *gyrA* in all isolates. *P*_PROB_ and mean concentrations corresponding to susceptible and QR *Salmonella* contaminated in the pork samples are shown in Table 2. The *P*_E_ to susceptible and QR *Salmonella*-contaminated pork at retail in Chiang Mai province was in the range of 2 × 10^−7^–0.03 (Table 3).

### 3.2. Hazard Characterization

The P_I_ and P_MI_ from QR Salmonella-contaminated pork at retails in Chiang Mai province were in the range of 2.2 × 10^−8^–3.1 × 10^−4^ and 3.9 ×10^−10^–5.4 × 10^−6^, respectively, while those from susceptible Salmonella-contaminated pork at retails were in the range of 1.8 × 10^−4^–3.2 ×10^−4^ and 2.3 × 10^−7^–4.2 × 10^−7^, respectively (Table 3).

### 3.3. Risk Characterization

The descriptive statistics and probability distributions of risk estimates in terms of *P*_FM_ and *M*_AFM_ from consuming retail pork contaminated with susceptible and QR *Salmonella* in Chiang Mai province, after performing a Monte Carlo simulation, are shown in Table 4 and Figure 1, Figure 2 and Figure 3. The mean *P*_FM_ of susceptible *Salmonella* was lower than that of QR *Salmonella* from the fresh market. On the other hand, in the modern trade, the mean *P*_FM_ of susceptible *Salmonella* became higher than that of QR *Salmonella*, essentially because the mean concentration of susceptible *Salmonella* was much higher than that of QR *Salmonella*.

## 4. Discussion

Two major approaches to AMR RA were determined by the data characteristics. The qualitative approach requires only a few calculations. The data variable is measured by the ordinal scale, e.g., low, moderate, and high. This could avoid complicated mathematical models and statistics, thus rendering risk assessment more straightforward, prolific, and time-saving. Nevertheless, the major drawback of qualitative AMR RA is the inherent subjectivity. One recommended solution to this dilemma is to transparently state or match the numerical values corresponding to individual descriptive terms for a qualitative variable [38,39]. Even though CAC encourages the quantitative technique to be performed as much as possible, the qualitative technique could not be discounted [4]. For the “quantitative technique”, the variables are measured by either interval or ratio scale. Two subcategories of quantitative AMR RA are deterministic and stochastic methods. Variables in the deterministic method possess only one single value, while those in the stochastic method encompass probability density corresponding to all possible values of a variable in the form of probability distribution [40,41,42]. This technique is more objective than the former technique, while complicated mathematical models are involved in almost every step of AMR RA (from hazard characterization to risk characterization) since the data in this study were allowed to quantitatively evaluate the mortality risk using Monte Carlo simulations. Therefore, the outputs from the mathematical models such as *P*_E_, *P*_MI_, and risk estimate are comparable whether between susceptible and QR *Salmonella* or fresh market and modern trade.

To better quantify the risk of exposure to the hazard, the types of hazard should be defined. Hazard, in the context of AMR RA, is either AMR pathogenic bacteria or an AMR determinant. The former hazard or sometimes so-called direct hazard in food is the AMR pathogenic microorganism being capable of colonizing and then infecting a human host. Furthermore, the direct hazard is also derived from handling contaminated food [43], while AMR bacteria harboring resistance genes directly transfer resistance genes to pathogenic bacteria or indirectly transfer to the commensal bacteria. The AMR determinant or resistant genes transferred through the last two mechanisms is a so-called indirect hazard [4]. This study determines the AMR hazard by both phenotypic and genotypic analyses; therefore, the AMR *P*_PROB_ is more conservative and prevalent than taking into account only the AMR hazard from the genotypic analysis [44].

This study collected pork samples in Chiang Mai province in northern Thailand to investigate the risk of consuming pork contaminated with susceptible and QR *Salmonella.* The *P*_PROB_ of susceptible and QR *Salmonella* isolated from the fresh market were in the narrow range of 28–30% (Table 2), while the *P*_PROB_ of susceptible *Salmonella* was about 10 times higher than the *P*_PROB_ of QR *Salmonella* isolated from the modern trade. The overall *P*_PROB_ of (both susceptible and QR) *Salmonella* from the fresh market is eight times more than the *P*_PROB_ of *Salmonella* from the modern trade. Likewise, QR *Salmonella* from the fresh market is almost 15 times more prevalent than susceptible *Salmonella* from the modern trade. In 2014, a similar study collected pork samples to compare the prevalence of susceptible and AMR *Salmonella* from the fresh market and the modern trade in Chiang Mai [45]. Even though 73% of fresh-market pork contaminated with *Salmonella* was more prevalent than only 10% of modern-trade pork contaminated with *Salmonella*, *Salmonella* prevalence from the fresh market in this previous study was slightly higher than the *P*_PROB_ of *Salmonella* from the fresh market in our study. These compatible findings suggest that the sanitation along the pork supply chain of the fresh market in Chiang Mai province should have been improved.

Even though several *Salmonella* contaminations along the pork supply chain from farms and slaughterhouses to retail were reported in Chiang Mai province in northern Thailand [11,46,47,48,49,50], the magnitude of the contamination of *Salmonella* was reported as a percentage by the detection technique, since the risk assessment approach recommended by the Codex Alimentarius requires both the prevalence and concentration of *Salmonella*, particularly at the point of consumption. Only one previous study in Chiang Mai reported that *Salmonella* prevalence and concentration in pork from the fresh market were 39% (27/70) and 1.31 ± 0.25 log MPN/g, respectively [46]. The mean concentration of *Salmonella* from the previous study was lower than that of *Salmonella* from the fresh market in our study at 1.8 ± 0.8 log cfu/g (Table 2). We assume that MPN/g and cfu/g are compatible units and take into account the standard deviations from these two studies; so far *Salmonella* concentration in pork from the fresh market has never been changed. Note that the Commission Regulation on the microbiological criteria for foodstuffs indicated that *Salmonella* was not detected in the area tested per pig carcass after dressing but before chilling by the EN/ISO 6579 analytical reference method [50].

In this study, *P*_E_ as a result of the exposure assessment step was derived from *P*_PROB_ and the concentration of either susceptible or QR *Salmonella*, including the pork consumption of the Thai population, as shown in Equation (7) [6]. An alternative model to determine human exposure to AMR hazards per person per day requires additional parameters such as cross-contamination, which is dependent upon transfer rates between the food product and the environment [51]. The *P*_E_ of QR *Salmonella* from fresh-market pork is considered low at 3 × 10^−2^, while the *P*_E_ of QR *Salmonella* from modern-trade pork at 2 × 10^−7^ is considered negligible [52]. These results indicate that the *P*_E_ of QR *Salmonella* from fresh market and modern trade followed the magnitude of both *P*_PROB_ and the concentration of QR *Salmonella*.

In terms of hazard characterization, the consequence of hazard was determined by the dose–response model while AMR RA additionally includes the consequence of AMR [4] as the probability of mortality given illness (*P*_MI_) in this study. The *P*_MI_ of QR *Salmonella* in the fresh market is much higher than *P*_MI_ in the modern trade (Table 3), primarily because the probability of exposure (*P*_E_) of QR *Salmonella* in the fresh market is higher than the *P*_E_ in the modern trade. In general, the *P*_E_ model is determined by *P*_PROB_ and the concentration (*C*_S_) of *Salmonella* (Equation (7)). This indicates that the adverse health effect of QR *Salmonella* from consuming fresh-market pork was higher than that from consuming modern-trade pork in Chiang Mai province.

So far, there have been very few risk assessments evaluating human health effects due to AMR *Salmonella*. One of these studies was the risk assessment of AMR *Salmonella* related to cattle [53,54]. A qualitative approach evaluated the additional risk of QR *Salmonella* recovered from minced pork as high [25]. Another qualitative risk assessment of human health effects from QR *Salmonella* Typhimurium in the EU upon using a (fluoro)quinolone in livestock (not necessarily swine) suggested the risk was low [55]. However, a quantitative risk assessment evaluated the human health effects of multi-resistant *Salmonella* Typhimurium DT104-contaminated Danish pork sausage [24]. The risk of salmonellosis from consuming such dry-cured pork sausages was in the range of 2.5 × 10^−8^–1.9 × 10^−6^, whereas in our study the mean mortality risks of QR *Salmonella* from modern-trade and fresh-market pork were as low as 7.4 × 10^−17^ and 2.0 × 10^−7^, respectively.

A previous study in Thailand reported that the annual mean mortality rate in 2009 (calculated from an average of the annual mortality cases of four major AMR bacteria (*Acinetobacter baumannii, Staphylococcus aureus, Klebsiella pneumoniae*, and *E. coli*)) was about 14.8 per 100,000 Thai population and was assumed to be the annual mean mortality rate for AMR salmonellosis [3]. In this study, the annual mortality rates for QR salmonellosis simulated by the risk assessment models were in the range of 0–32, which is in line with a previous study. The risk assessment models used in both exposure assessment and hazard characterization were applicable to evaluate the adverse health effects of AMR *Salmonella* in Thailand.

## 5. Conclusions

As far as we are aware, this is the first study of the quantitative microbial risk assessment of QR *Salmonella* in retail pork using a Monte Carlo simulation to comparatively report the human health adverse effects of susceptible and QR *Salmonella* from consuming retail pork from fresh market and modern trade, particularly in Thailand. The *P*_PROB_ of both susceptible and QR *Salmonella* from the retail market are higher than the *P*_PROB_ from modern trade. Likewise, the risk estimate in terms of the annual mortality rate of QR *Salmonella* from the fresh market is higher than that of QR *Salmonella* from modern trade and is also in line with a previous study reporting the mortality rate of AMR pathogens. The risk assessment models used in this study fit for evaluating the adverse health effects of QR *Salmonella* in Thailand and that of other foodborne AMR pathogens.

## Figures and Tables

**Figure 1 foods-11-02942-f001:**
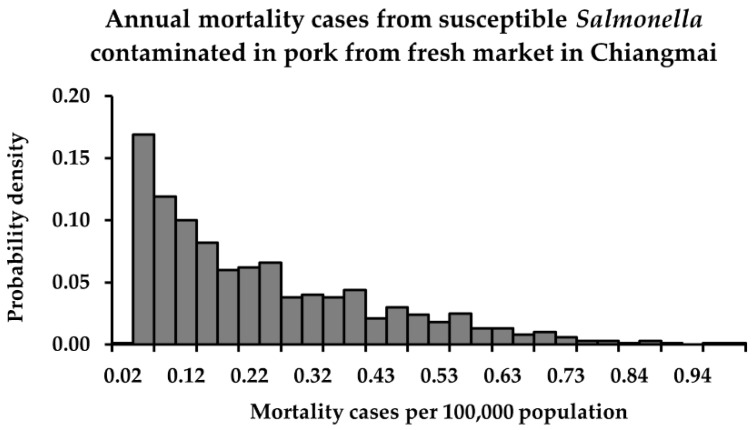
Annual mortality cases from susceptible *Salmonella*-contaminated pork from the fresh market in Chiang Mai.

**Figure 2 foods-11-02942-f002:**
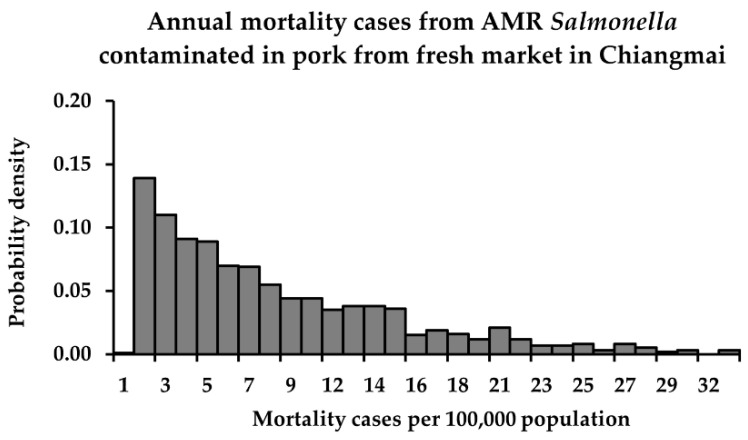
Annual mortality cases from QR *Salmonella*-contaminated pork from the fresh market in Chiang Mai.

**Figure 3 foods-11-02942-f003:**
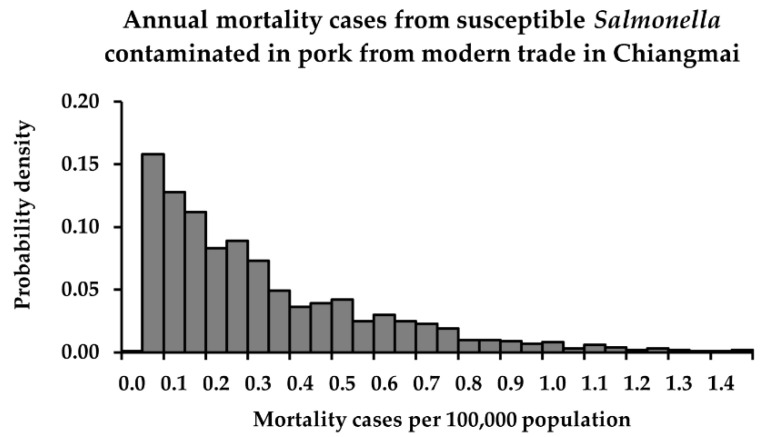
Annual mortality cases from susceptible *Salmonella*-contaminated pork from the modern trade in Chiang Mai.

**Table 1 foods-11-02942-t001:** No. of *Salmonella* positive samples collected from retailers in Chiang Mai province.

Retail	No. of *Salmonella*		Total
	Susceptible	QR	
Fresh market	30	28	58 (*n* =100)
Modern trade	6	0	6 (*n* = 50)

**Table 2 foods-11-02942-t002:** *P*_PROB_ and mean concentration of contaminants in the pork samples.

Retail	*P*_PROB_ (%)			Mean Concentration ± SD (log cfu/g)		
	*Salmonella* spp.		Total	*Salmonella* spp.		Total *
	Susceptible	QR		Susceptible	QR	
Fresh market	30.4	28.4	57.8	1.5 ± 0.8	2.1 ± 0.7	1.8 ± 0.8
Modern trade	13.5	1.9	6.9	1.9 ± 0.9	0	1.9 ± 0.9

* Accounted for only positive samples.

**Table 3 foods-11-02942-t003:** Probabilities of exposure (*P*_E_), illness (*P*_I_), and mortality given illness (*P*_MI_) from susceptible and QR *Salmonella* spp.

Retail	*P* _E_		*P* _I_		*P* _MI_	
	*Salmonella* spp.		*Salmonella* spp.		*Salmonella*	
	Susceptible	QR	Susceptible	QR	Susceptible	QR
Fresh market	0.020	0.030	1.8 × 10^−4^	3.1 × 10^−4^	2.3 × 10^−7^	5.4 × 10^−6^
Modern trade	0.016	2 × 10^−7^	3.2 × 10^−4^	2.2 × 10^−8^	4.2 × 10^−7^	3.9 × 10^−10^

**Table 4 foods-11-02942-t004:** Descriptive statistics of risk estimate (*P*_FM_) and annual mortality rate (*M*_AFM_) from consuming pork contaminated with susceptible and AMR *Salmonella* spp.

Retail		Risk Estimate		Annual Cases *	
		*Salmonella* spp.		*Salmonella* spp.	
		Susceptible	AMR	Susceptible	AMR
Fresh market	min	5.3 × 10^−13^	8.8 × 10^−11^	<1	<1
	mean	5.7 × 10^−9^	2.0 × 10^−7^	<1 *^a^*	7 *^b^*
	max	2.7 × 10^−8^	8.8 × 10^−7^	1	32
Modern trade	min	1.5 × 10^−12^	4.2 × 10^−21^	<1	<1
	mean	7.9 × 10^−9^	7.4 × 10^−17^	<1 *^c^*	<1 *^d^*
	max	4.0 × 10^−8^	7.6 × 10^−16^	2	<1

* Mean annual cases per 100,000 population (*P*_AFM_) with different letters implies that there are statistically significant differences (*p* < 0.05) (letters *a* through *d*).

## Data Availability

All the data presented within the article is available upon request from the corresponding author.

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
