# Peer review of "Quantitative Risk Assessment of Susceptible and Ciprofloxacin-Resistant Salmonella from Retail Pork in Chiang Mai Province in Northern Thailand"

_foods, 2022, doi:10.3390/foods11192942_

Round 1

Reviewer 1 Report

Although the topic is interesting,  the paper is not suitable for publication in its present form.  Some of the revisions are listed below, and please check all manuscript

In the Introduction section, the authors stated that " Even though 34 AMR RA relating to retail foods have been reported, only 8 articles investigated the adverse health effects of AMR Salmonella spp. Only half of these reports are related to the pork supply chain " .  They stated this information depends on a study in 2019. It should be updated and all references stated or this part should be rewritten. 

In the methodology, general microbiological analyzing methods (MPN) are explained in detail. It should be summarized.  On the other hand, methods used in antimicrobial susceptibility testing and the determination of antimicrobial resistance genes should be explained in more detail. 

The links between paragraphs in the manuscript can be edited a little better Results and Discussion is not clear. It is not well organized. It should be rewritten. In the manuscript, figures 1-3 are missing. 

The final conclusions should include key findings and more specific.

There are also grammatical errors and unclear sentences in the manuscript.  Extensive editing of English language is required.

Reviewer 2 Report

Dear Prof. Dr.

Editor of Foods

Hope this email finds you well.

I have read carefully the manuscript entitled "Quantitative risk assessment of susceptible and ciprofloxacin resistant Salmonella from retail pork in Chiang Mai province in

northern Thailand".

·       I believe the work done in this manuscript is valuable enough to continue studying, however, this manuscript should improve several points before it will be considered:

Comments:

·       The cited references in text should be untied, the authors in general and according to journal instructions cited references in text in numbers however, through the text they used names for example page 4 of 12 paragraph 2.4 Determination of antimicrobial resistance genes.   …… the plasmid-mediated quinolone resistance genes (Ciesielczuk, Hornsey et al. 2013, Lu, Zhao et al. 2015). Change to numbers.

·       The abbreviation should be defined in the abstract and through the text.

·       The abstract captures the main points of the paper and should contain the spot point of obtained results.

·       The manuscript is preceded by 3 to 6 keywords in alphabetic order.

·       Some crucial and related previous researches of antimicrobial resistance were not included in the introduction section.

·       The strategy of quantitative risk assessment mechanism should be described more clearly.

·       The statistic evaluation of the assessment should be performed.

·       The authors used many types of fonts as well different font size please check the text again just example page 2 of 12 the paragraph no. 4.

·       The manuscript's English should be checked typing and grammatically seriously.

·       The authors have a big issue with the references and should be check again just few examples:

·       References must be checked (Unity) Authors names, year, title, volume, page number. In some references the journal names are abbreviated in other ones not like in ref. no. 3, 5, 35, and 52 journal names in full names, where other are abbreviated.

·       References must be updated.

·       Ref. no. 1 check the authors names.

·       Ref. no. 4 and 5 check again.

·       Many ref. miss the Journal names like ref. no. 9, 23, 25 just for example

·       Many ref. miss the volume or page numbers or both like ref. no.11

The work idea, is neatly fine except the mistyping and the past comments written before. The paper is accepted to publish with major correction and carful written.

With my regards     

Round 2

Reviewer 1 Report

Although an editing service edited the manuscript, wrong writings of words ( for example, in the newly added paragraph in Discussion and other parts of the text) are available. 

The paragraphs in the Discussion section should be linked to each other to increase the readability of the manuscript.  

Reviewer 2 Report

English language and style are fine/minor spell check required
